# Molecular Characterization of Three B-1,4-Endoglucanase Genes in *Pratylenchus loosi* and Functional Analysis of *Pl-eng*-2 Gene

**DOI:** 10.3390/plants10030568

**Published:** 2021-03-17

**Authors:** Negin Mirghasemi, Elena Fanelli, Salar Jamali, Mohammed Mehdi Sohani, Francesca De Luca

**Affiliations:** 1Plant Protection Department, Faculty of Agricultural Sciences, University of Guilan, Rasht 4199613769, Iran; negin.mirghasemi54@gmail.com (N.M.); jamali@guila.ac.ir (S.J.); 2Istituto per la Protezione Sostenibile delle Piante (IPSP), Consiglio Nazionale delle Ricerche (CNR), S.S. Bari, Via G. Amendola 122/D, 70126 Bari, Italy; elena.fanelli@ipsp.cnr.it; 3Department of Biotechnology, Faculty of Agricultural Sciences, University of Guilan, Rasht 4199613769, Iran; msohani@guilan.ac.ir

**Keywords:** cellulase, evolution, gene duplication, intron, RNA interference, root-lesion nematode

## Abstract

*Pratylenchus loosi* is an important root-lesion nematode that causes damage to tea plantations in Iran and all over the world. The present study reports on the characterization and evolution of three ß-1,4-endoglucanase genes: *Pl-eng*-2, *Pl-eng*-3 and *Pl-eng*-4. The gene structure of *Pl-eng*-2 was fully determined with the predicted signal peptide and devoid of the linker domain and carbohydrate-binding domain, while *Pl-eng*-3 and *Pl-eng*-4 were only partially sequenced. The transcription of *Pl-eng*-2 was localized in the secretory esophageal glands of all life stages, but it was upregulated in male and female stages. The exon/intron structures of *Pl-eng*-2, *Pl-eng*-3 and *Pl-eng*-4 confirmed that they resulted from gene duplication followed by sequence and gene structure diversification with loss of the linker domain and carbohydrate-binding domain during evolution. A phylogenetic analysis further confirmed that nematode endoglucanases resulted from the horizontal gene transfer of a bacterial gene, as *Pl-eng*-3 showed sister relationships with the *Cel*B cellulase of *Bacillus subtilis*. Silencing *Pl-eng*-2 by in vitro RNA interference produced a 60% decrease of the transcript level. The reproductive ability of silenced *P. loosi* showed a 35% reduction of eggs and larval stages compared to untreated nematodes, suggesting that this gene is involved in the early steps of invasion.

## 1. Introduction

Root-lesion nematodes (RLN) belonging to the genus *Pratylenchus* Filipjev 1936 [1] consist of about 100 species identified so far and are considered as one of the most devastating migratory nematodes, along with root-knot and cyst nematodes [2,3,4]. *Pratylenchus* spp. are widely distributed in cool, temperate and tropical environments, thus resulting in being highly relevant to agriculture [5]. All life stages can penetrate the root, freely move inside the root, feed and destroy tissues through the combination of mechanical action and secretion of cell wall-modifying enzymes (CWMPs). These nematode secreted proteins are present in plant parasitic nematodes and play an important role in plant-nematode interactions, allowing the penetration, migration and evasion of host defenses [6,7,8,9,10,11,12,13]. In plant parasitic nematodes, these enzymes have been acquired by horizontal gene transfer (HGT) from bacteria or fungi [14,15]. So far, a few complete endoglucanase genes (*engs*), along with many partial sequences from *Pratylenchus* species, have been reported [9,11,12,13,16,17]. No information on the cell wall-modifying enzymes in the tea nematode *Pratylenchus loosi* Loof 1960 [18] is reported in the literature. *Pratylenchus loosi* is considered a serious nematode pest causing yield losses in tea plantations in Iran and all over the world [19]. The present study reports on the isolation of three fragments coding for *engs*, named *Pl-eng*-2, *Pl-eng*-3 and *Pl-eng*-4, and the functional characterization of *Pl-eng*-2. Phylogenetic analyses revealed the occurrence of multiple genes that underwent, after duplication, rapid diversification, as shown by the position of *Pl*-Engs in different groups.

## 2. Results

### 2.1. Molecular Characterization of Pl-engs in Pratylenchus loosi

PCR amplification performed on the genomic DNA of *P. loosi* using ENG1 and ENG2-degenerated primers [5] generated a product of about 460 bp. This fragment was cloned and sequenced. Sequence analyses of several clones revealed the presence of three different fragments of 460 bp, 427 bp and 389 bp in length. BLASTX analyses revealed that all three fragments showed similarities with *Pratylenchus* endoglucanases—in particular, the 460-bp fragment showed 90% similarity (275/306 identities and 287/306 positives) with *P. coffeae* endoglucanase and 83-84% with *Pv-eng*-2 and *Pv-eng*-1 of *P. vulnus*. The 427-bp fragment showed 60% similarity with *Meloidogyne* endoglucanases; the 389-bp fragment showed 53% similarity with *Ditylenchus destructor* and 51% with *Meloidogyne incognita*. Based on the similarity with the *engs* available in the database, the three different fragments were named *Pl-eng*-2, *Pl-eng*-3 and *Pl-eng*-4. The 460-bp fragment of *Pl-eng*-2 showed one intron and differed from the 427-bp fragment of *Pl-eng*-3 for the presence of an intron in a different position, while no intron was present in the 389-bp fragment of *Pl-eng*-4. Pairwise comparisons between the three partial *eng* sequences revealed a 59% similarity between *Pl-eng*-2 and *Pl-eng*-3 and 48% with the intron-less *Pl-eng*-4, while *Pl-eng*-3 and *Pl-eng*-4 showed a 54% similarity.

The full-length cDNA of *Pl-eng*-2 was obtained by 3′/5′ RACE experiments and contained 1152 nucleotides (excluding the poly (dA) tail) with an open reading frame (ORF) of 984 bp.

The cDNA contained a 16 bp 5′ UTR and a 152 bp 3′ UTR, which encompassed the cytoplasmic polyadenylation element (CPE; TTTTTAT) located upstream from the polyadenylation signal (AATAAA) and signals (TAAAT) involved in the regulation of stability and translation at the mRNA level. The ORF encoded a deduced protein of 327 amino acids with a calculated molecular weight of 35.37 kDa and a pI of 8.21. A secretion signal sequence terminated immediately upstream of a protease cleavage site between amino acids Gly21 and Ala22 and was predicted by the Signal P 4.0 World Wide Web Server [20]. No putative N-glycosylation site was present in the sequence.

The enzyme is composed of a catalytic domain, but the linker and carbohydrate-binding module (CBM) are not encompassed. A glycoside hydrolases family 5 signature extends from residues 150 to 159.

### 2.2. Gene Structure

The amplification of *P. loosi* gDNA, using primers located in the UTR regions for cDNA, produced a fragment of 1065 bp for *Pl-eng*-2.

The alignment of the *Pl-eng*-2 genomic sequence with the corresponding cDNA sequence revealed the presence of only two exons and one intron. The intron size is 69 bp, and the intron–exon junctions GT and AG were conserved. The intron size of *Pl-eng*-3 is 32 bp, and the intron–exon junctions GT and AG were also conserved.

The comparison of the gene structure of *Pl-eng*-2 with the partial *Pl-eng*-3 and *Pl-eng*-4 from *P. loosi*, along with the *Pratylenchus* spp. *engs* available in the database (Figure 1), revealed that the intron position of *Pl-eng*-2 is conserved among *Pratylenchus* spp. [11,12], in contrast with the intron position of *Pl-eng*-3, which is only conserved in a few *Pratylenchus engs* present in the database (Figure 1).

### 2.3. Phylogenetic Analysis

The phylogenetic analysis revealed that *Pl-eng*-2, *Pl-eng*-3 and *Pl-eng*-4 proteins were grouped into three separate subgroupings (Figure 2) according to their evolutionary relationships. *Pl*- *eng*-2 was closely related to *P. coffeae* eng-1 and *P. vulnus eng*-8 and, all together, clustered in a large group containing *Pl*-*eng*-3 and *Pl*-*eng*-4 from *P. loosi* and most of the *Pratylenchus* engs, along with several *Meloidogyne*, *Radopholus, Aphelenchoides avenae*, *Globodera rostochiensis* and *Heterodera glycines* engs. Furthermore, the *Pl*-*eng*-3 results were closely related to *CelB*-*eng* of *B. subtilis* and *G. rostochiensis eng*-3 and *eng*-4 and *H. glycines eng*-5, while the *Pl*-*eng*-4 results were closely related with *A. besseyi eng*-1, *Pv*-*eng*-2, *Rr*-*eng*-1, *Hg*-*eng*-6, *Pg*-*eng*-1 and *Ppr*-*eng*-2. *Bursaphelenchus xylophilus eng*-1 and *eng*-3, belonging to the GHF45 family, are positioned at the basal position of the phylogenetic tree. 

### 2.4. Expression Profile by Real Time-PCR (qRT-PCR)

Expression profile of *Pl-eng*-2 in different developmental stages was analyzed by using gene-specific primers in qRT-PCR on the total RNA from juveniles and adult females and males. The transcript levels of *Pl-eng*-2 were highest in the adult males (eight-fold) and females (seven-fold) compared to the juveniles (Figure 3).

### 2.5. In Situ Localization of Pl-eng-2 Transcript 

The tissue localization of the *Pl-eng*-2 transcription in the *P. loosi* mixed stages was analyzed by in-situ hybridization on the fixed nematode sections. The antisense *Pl-eng*-2 probe specifically hybridized in the esophageal gland cells of the nematodes (Figure 4A), whereas no signal was observed with the control sense probe (Figure 4B).

### 2.6. Silencing of Pl-eng-2

The effect of RNAi silencing was detected by quantitative PCR methods (qPCR) after soaking *P. loosi* nematodes for 24h in *Pl-eng*-2 dsRNA. A specific region of the 18S rRNA gene was used as an endogenous reference for normalization. The relative expression of *Pl-eng*-2 in the nematodes soaked in *Pl-eng*-2 dsRNA was compared with the relative expression of *Pl-eng*-2 in the control nematodes soaked in the soaking buffer and in *gfp* dsRNA.

The expression of *Pl-eng*-2 in nematodes treated with *Pl-eng-2* dsRNA decreased by 60% (*p* < 0.01) compared with the untreated nematodes (Figure 4). Such a decrease in *Pl-eng*-2 expression was more evident when nematodes treated with *Pl-eng*-2 dsRNA were compared with *gfp* dsRNA-treated nematodes (83%) (Figure 5).

The phenotype effect of gene silencing on the nematodes was determined. Nematodes phenotype observation with a microscope after nematode soaking in *Pl-eng-2* dsRNA revealed the phenomenon of a straight shape and behavioral aberration after 24-h incubation. This phenomenon was less evident after nematode soaking in *gfp* dsRNA (Figure 6).

The effect of *Pl-eng-2* silencing in *P. loosi* was studied, and the ability of nematodes to reproduce was determined after transferring untreated and dsRNA-treated nematodes on mini-carrot discs. The nematode progenies were counted and compared 45 days after inoculation.

The total number of eggs and larval stages after 45 days showed a reduction (*p* < 0.05) of 35% (Figure 7) compared with that retrieved from discs infected with untreated nematodes. No differences were observed in the number of males and females (Figure 7). 

## 3. Discussion

The current study reports on the characterization of *Pl-eng*-2 gene in the tea nematode *P. loosi*, along with two partial sequences named *Pl-eng*-3 and *Pl-eng*-4, all encoding ß-1,4 endoglucanases. The *Pl-eng*-2 fragment was fully characterized at the genomic and cDNA levels, revealing the presence of a signal peptide for secretion and the catalytic domain directly joined to the 3′UTR region. The 3′UTR of *Pl-eng*-2 is 152 bp in length, similar to those of other nematode *engs* without a linker region and cellulose-binding domain but shorter compared to those of *engs* with a linker and cellulose-binding domain. It is well-known that the length of the 3′UTR region is strictly related with mRNA stability, gene expression and cellular proliferation. Thus, shorter 3′UTR are more stable, with a higher transcription capacity and higher potential for protein production. This finding further demonstrates the important role of the 3′UTR of *engs* without a carbohydrate-binding domain during parasitism, allowing nematodes to quickly adapt to specific host plants. 

The *Pl-eng*-2, *Pl-eng*-3 and *Pl-eng*-4 genes in *P. loosi* showed high sequence dissimilarities each other and, also, gene structures. Both *Pl-eng*-2 and partial *Pl-eng*-3 sequences contained one intron differing in sequence and position, while the partial *Pl-eng*-4 sequence showed no intron (Figure 1). The single intron of *Pl-eng*-2 was in a conserved position in most of *eng* genes of *Pratylenchus* spp., suggesting that this intron was already present in the corresponding ancestral gene before the duplication event occurred. Furthermore, the existence of multiple genes for endoglucanases and the retention of duplicated genes in root-lesion nematodes may be correlated to the ability of these nematodes to freely move inside the roots and to parasitize a large number of plant hosts [11,12,13,17,21]. A phylogenetic analysis using the GHF5 catalytic cellulase domains of plant parasitic nematodes revealed that *P. loosi* ENGs grouped into different subgroupings (Figure 2), confirming that, after the HGT event, the gene was immediately duplicated, and sequence diversification occurred even if the function was maintained. *Pl-*e*ng*-2 showed sister relationships in the phylogenetic tree based on the catalytic cellulose domains with other nematode ENGs with and without CBM, confirming that most of the nematode ENGs have lost this domain during evolution in order to quickly adapt to new host plants. Moreover, the close relationships of *Pl-eng*-3 with *Cel*B of *B. subtilis* and the occurrence of *Pl-eng*-2 in the same large grouping with other nematode ENGS confirms that *Pratylenchus engs* were also acquired from bacteria by horizontal gene transfer. The phylogenetic data and the different gene organization of *Pl*-*engs* demonstrated that, immediately after duplication, the ancestral gene underwent diversification, as shown by different intron positions between *Pl-eng*-2 and *Pl-eng*-3 and the absence of an intron in the *Pl-eng*-4 partial sequence. The conservation of the intron position of *Pl-eng*-2 in most *engs* of *Pratylenchus* (Figure 1) suggests that this intron may occur in the common ancestral gene. Interestingly, the intron position of the partial *Pl-eng*-3 is conserved in a few *Pratylenchus* species, confirming that the sequence and gene diversification immediately occurred after duplication.

*Pl-eng*-2 is expressed in all life stages of *P. loosi*, but its level is higher in adult stages, confirming the active role in both the male and female stages of *P. loosi* during parasitism, as also found in *P. vulnus* [11,12]. This finding was furtherly confirmed by the in-situ localization of *Pl-eng*-2 in the pharyngeal gland cells both in females and males. To prove further the role of *Pl-eng*-2 during parasitism, the gene was knocked down by using RNAi [12,22,23,24]. Silenced nematodes, after 24 h of incubation, showed a 60% reduction of *Pl-eng*-2 expression by using qRT-PCR (Figure 5) and a straight shape at a microscopy observation (Figure 6), confirming the effect of dsRNA. Then, untreated and silenced nematodes were incubated on carrot discs, and, after 45 days, a significant reduction of eggs and larval stages (35%) was observed compared to the control (Figure 7), while the ratio between males and females was identical to that of the nematode control (Figure 7). This result strongly suggests that the silencing effect of *Pl-eng*-2 is not durable and the involvement of *Pl-eng*-2 in the migration and feeding of adult stages inside the roots, as well as nematode reproduction. In conclusion, our data demonstrated that *P. loosi* also contains several *eng* genes, such as other *Pratylenchus* species. Furthermore, our observation that most of *Pratylenchus* effectors without CBM contain shorter 3′UTR region allows us to speculate that both features are needed to attack several host plants and to quickly adapt to them. Finally, this basic knowledge on *Pratylenchus eng*s can contribute to more insights into how to develop novel molecular strategies to control RLN.

## 4. Materials and Methods

### 4.1. Nematode Collection

A *P. loosi* population from Iran was isolated from tea plant roots and, starting from single females, reared on sterile carrot discs. Every two months, mixed stage nematodes, recovered from carrot cultures, were sterilized with 0.02% ethoxyethyl mercury and 0.1% streptomycin sulphate solutions for 2 and 24 h, respectively, and rinsed twice with sterilized water. Under aseptic conditions, fresh carrots were treated with 20% NaOCl solution for 5 min, flamed and then peeled and sliced transversely (10-mm thick, 30–40-mm diameter). Sterilized nematodes (100–200) were transferred onto a single carrot disc into a sterile 50-mm Petri dish and then grown at 23 °C in the dark for up to 8 weeks [11,25]. Nematodes were recovered from carrot discs by cutting the discs into smaller pieces with a sharp sterile scalpel and submerging in water in Petri dishes. The dishes were then incubated at room temperature for 24 h. The suspended nematodes were collected by sterile Pasteur pipettes, cleaned by repeated washes with sterile water and decanted into a glass beaker. The nematodes were counted under a dissecting microscope.

### 4.2. DNA and RNA Isolation

Total genomic DNA and RNA of *P. loosi* mixed life stages were extracted using AllPrep DNA/RNA kit (QIAGEN, Hilden, Germany) according to the manufacturer’s instructions. Genomic DNA and total RNA were quantified using Nanodrop.

### 4.3. Genomic DNA Amplification

Degenerate primer set (ENG1/ENG2 [6].) were used to amplify the conserved catalytic domain for endoglucanases. PCR assay was conducted as described by Fanelli et al., [11].

Cycling conditions used were: an initial denaturation at 94 °C for 2 min, followed by 35 cycles of denaturation at 94 °C for 20 s; annealing at 65 °C for 5 s, 60 °C for 5 s, 55 °C for 5 s, 50 °C for 5 s and extension at 68 °C for 1 min and a final step at 68 °C for 15 min. PCR products were gel purified using the protocol listed by the manufacturer (NucleoSpin Gel and PCR Clean-up, Machery Nagel, Dueren, Germany) and, subsequently, ligated into the pGEM T-easy vector (Promega, Madison, Wisconsin, USA) and used to transform JM109 (Promega, Kilkenny, Ireland) chemically competent *Escherichia coli* cells, which were spread on LB-agar plates with ampicillin and grown overnight at 37 °C. Three insert-positive clones were grown in 3 mL of LB overnight at 37 °C, followed by plasmid DNA extraction (QIAprep Spin Miniprep kit; QIAGEN, Hilden, Germany), and sequenced by Eurofins (Rosenow, Germany). Sequences were analyzed by the BLAST tool at NCBI.

### 4.4. Rapid Amplification of cDNA Ends (RACE)

The 3′/5′ RACE of *Pl-eng*-2 was carried out on 1 µg of total RNA from nematode mixed stages with SMARTer^®^ RACE 5′/3′ (Clontech, California, USA) according to the manufacturer’s instructions. The 3′ end of *Pl-eng*-2 was amplified using the gene specific primer 3′RACE *Pl*5 *eng* 3 (CGTGGATGTGGTGGCGGCGA) and a long universal primer UPM.

Nested PCR, using the short universal primer UPS and 3′RACE *Pl*5 *eng* 2 (GCCAGTGTTGTGAAGCCATACC) specific primers, generated a band that was cloned and sequenced.

The 5′ end of *Pl-eng*-2 was generated using the gene specific primer 5′RACE *Pl*5 *eng* 3 (TCGCCGCCACCACATCCACG) and a long UPM. Nested PCR, using the short universal primer UPS and gene specific primers 5′RACE *Pl*5 *eng* 2 (GGTATGGCTTCACAACACTGGC), generated a band that was cloned and sequenced. 5′/3′ PCR reactions were performed as follows: 94 °C for 2 min; 30 cycles at 94 °C for 30 sec, 66 °C for 30 s, and 72 °C for 3 min and a final extension step at 72 °C for 7 min.

### 4.5. Sequence Analysis

The nucleotide and translated amino acid sequences were analyzed for similarities to other genes and proteins using BLAST analyses against the NCBI nonredundant nucleotide and protein databases (http://www.ncbi.nlm.nih.gov/, accessed on 17 March 2021). Furthermore, protein sequence analyses were conducted using SIGNALP v. 4.0 to predict protein signal peptide [20] and ExPASy, the bioinformatics resource portal, to predict the protein molecular mass and the theoretical isoelectric point.

### 4.6. Phylogenetic Analysis

Protein sequences of catalytic cellulose domains of nematode ENGs were retrieved from the GenBank database according to their accession numbers. Seventy-two sequences, including several clones of *Pl-eng*-2, *Pl-eng*-3 and *Pl-eng*-4 from *P. loosi*, were aligned using MAFFT [26]. The final alignment was checked manually to correct the potential inconsistencies. Sequence alignments were manually edited using BioEdit in order to improve the multi-alignment. Outgroup taxa, *B. subtilis* CelB [27] and *B. xylophilus eng*-1 and *eng*-3 [28], were chosen according to the results of previously published data. Phylogenetic trees were performed with the Maximum Likelihood (ML) method using MEGA package version X software [29]. The phylograms were bootstrapped 1000 times to assess the degree of support for the phylogenetic branching indicated by the optimal tree for each method.

### 4.7. Expression Pattern Analysis

To investigate the expression pattern of *Pl-eng*-2 transcript among the different developmental stages, RT-PCR experiments were performed with 100 Juveniles (J2) and 50 females and 50 males of *P. loosi* that were handpicked under the microscope. Total RNA was extracted using the RNeasy Tissue Mini Kit following the manufacturer’s instructions (Qiagen) and further treated with an RNase-free DNase I set (Qiagen) to eliminate any contaminating genomic DNA. First-strand cDNA was synthesized starting from 150 ng of total RNA using a QuantiTect Reverse transcription kit (QIAGEN, Hilden, Germany) following the manufacturer’s instructions.

The relative expression among the life stages was calculated by using the ΔΔCt method. A portion of 18S rRNA gene was used as the endogenous control using specific primers: q18SPlfor (AAGCCGACAATGAACCAGTAC) and q18SPl rev (ATGAGAGGGCAAGTCTGGTG).

Real-time PCR was performed in 25 µL volumes containing 10 ng of cDNA, 12.5 µl 2 × Fast Start SYBR Green master mix (Roche) and 10 pmol of each specific primer.

Gene-specific primers *Pl*engfor5 (GATTGGTGGACTTTCCTCGA) and *Pl*engrev3 (CGCATTTTTCACTCTGCTGCC) were used to determine the expression profile.

The thermal profile for real-time PCR was 10 min at 95 °C, followed by 40 cycles at 95° C for 30 s, 56 °C for 30 s and 72 °C for 40 s. Dissociation curve analysis of the amplification products was performed at the end of each PCR to confirm that only one PCR product was amplified and detected. The real-time experiments were conducted on a Stratagene thermal cycler, and fluorescent real-time PCR data were analyzed with MX3000P software.

### 4.8. In-Situ Hybridization

To assess the localization of the *Pl-eng*-2 transcript, whole-mount in-situ hybridizations were performed in all stages of *P. loosi* following the protocol of de Boer et al. (1998) [30].

For in-situ hybridization, oligonucleotides *Pl*_Eng_for3_int(GATTCGGTGCTGACTGCATC) and *Pl*_Engrev2_stop (CTCTGCTGCCCTTTTCAGCC) were used to synthesize a 100-bp fragment of *Pl-eng*-2. Digoxygenin-11-dUTP-labeled DNA probes were synthesized in sense and antisense directions using the PCR DIG Probe Synthesis kit (Roche Applied Science). Briefly, *P. loosi* mixed life stages were fixed in 2% paraformaldehyde for 18 h at 5 °C, followed by a second incubation for 4 h at room temperature. The nematodes were resuspended in 0.2% paraformaldehyde; cut into sections and permeabilized with proteinase K, acetone and methanol. The sections were hybridized overnight at 50 °C with the sense and antisense probes. The sections were washed three times at 50 °C in 4 × SSC and three times in 0.1 × SSC and 0.1% SDS. The hybridized probes within the nematode tissues were detected using anti-DIG antibody conjugated to alkaline phosphatase and its substrate. Nematode sections were then observed using a microscope.

### 4.9. RNAi of Pl-eng-2

Templates for dsRNA were made by PCR on cDNA generated from RNA extracted from mixed stages of *P. loosi* using a RNeasy kit (QIAGEN, Hilden, Germany) according to the manufacturer’s instructions.

Two separate PCR reactions were performed in which *Pl-eng*-2 was amplified with a T7 promoter sequence incorporated at the 5′ end of either the sense or antisense strand.

The *Pl-eng*-2 primer sequences used for the reactions were T7*Pl*Engfor5 (TAATACGACTCACTATAGGGGATTGGTGGACTTTCCTCGA) and *Pl*Engrev2stop (CTCTGCTGCCCTTTTCAGCC) in one reaction and *Pl*Engfor5 (GATTGGTGGACTTTCCTCGA) and T7 *Pl*Engrev2stop (TAATACGACTCACTATAGGG CTCTGCTGCCCTTTTCAGCC) in the second reaction.

Nonspecific control dsRNA (green fluorescent protein gene, *gfp*, 250 bp) was amplified by using specific primers GFPFOR (CACATGAAGCAGCACGACT) and GFPREV (GATATAGACGTTGTGGCTGT) from the cloning vector pA7-GFP.

PCR products were cleaned using a PCR purification kit (QIAGEN, Hilden, Germany), and 600–800 ng of each PCR product was used for in vitro transcription using a Megascript kit (Ambion, Huntingdon, England) according to the manufacturer’s instructions. The RNA generated in the two reactions was annealed to generate dsRNA. DNA and single-stranded RNA were removed by nuclease digestion (Megascript kit; Ambion, Huntingdon, England). dsRNA was purified using filter cartridges (Ambion, Huntingdon, England) and eluted in 50 µL elution solution (Ambion, Huntingdon, England). The dsRNA was verified by 1% agarose gel electrophoresis and quantified using a NanoDrop spectrophotometer. Two hundred *P. loosi* females and 100 males grown on carrot discs were both collected and soaked in 40 µL *Pl-eng*-2 dsRNA solution (1 µL/µL) for 24 h. Meanwhile, controls were incubated in either elution buffer or with *gfp* dsRNA. Treated and control nematodes were cleaned three times with DEPC-treated water, and total RNA was then extracted. qPCR was used to analyze the transcript suppression after RNAi treatment. All experiments were performed three times.

A specific portion of 18S rRNA gene was used as the endogenous control. Real-time PCR was performed in 25 µL volumes containing 10 ng of cDNA, 12.5 µL 2 × Fast Start SYBR Green master mix (Roche, Basel, Switzerland) and 10 pmol of each specific primer.

Gene-specific primers used to analyze the *Pl-eng*-2 transcript suppression were 3race*Pl5*Eng3 (CGTGGATGTGGTGGCGGCGA) and 5*Pl*Engrev4 (TTGTTCAACGCAGTTTGTGCC).

PCR was 10 min at 95 °C, followed by 40 cycles at 95 °C for 30 s, 58 °C for 30 s and 72 °C for 40 s. Dissociation curve analysis of the amplification products was also performed. 

### 4.10. Effect of Silencing on Reproduction of the Nematodes

The effect of gene silencing on nematode reproduction was determined. For the reproduction test, 30 treated and untreated nematodes (20 females and 10 males) were washed three times with sterile water and inoculated on carrot discs for 45 days, one life cycle of *P. loosi*. The nematodes were maintained at 23 ± 1°C. Each treatment was repeated 10 times. The number of *Pl-eng*-2 RNAi-treated and -untreated nematode progenies were counted and compared. The sum of the number of eggs, juveniles, females and males was considered as the final nematode population density (Pf) and was used to determine the reproduction factor.

## Figures and Tables

**Figure 1 plants-10-00568-f001:**
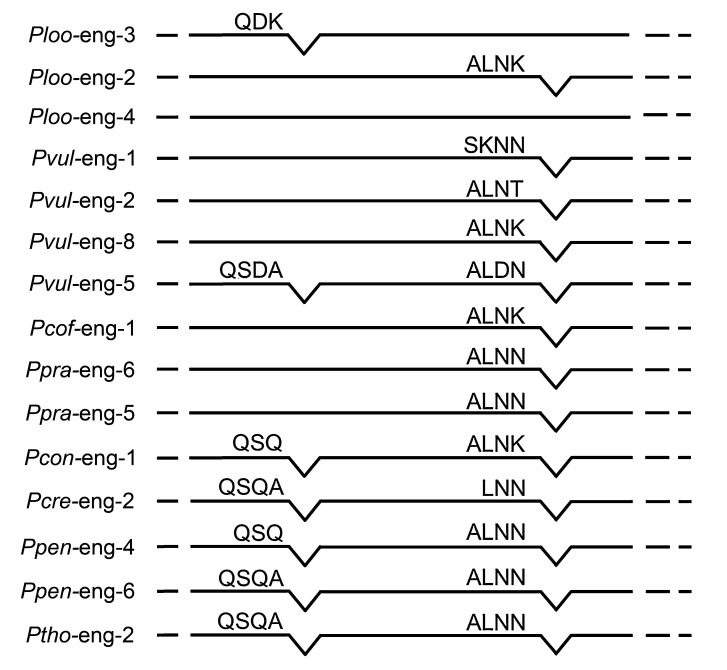
Schematic representation of the intron–exon structure of the *Pl-eng*-2, Pl-*eng*-3 and *Pl-eng*-4 genes from *P. loosi* compared with other *Pratylenchus* cellulase genes. Exons are shown as solid lines, and introns are shown as V-shaped lines. Solid V-shaped lines denoted introns located in conserved positions between nematode engs. The abbreviations indicate: Ploo: *P. loosi*, Pvul: *P. vulnus*, Pcof: *P. coffeae*, Ppra: *P. pratensis*, Pcon: *P. convallariae*, Pcre: *P. crenatus*, Ppen: *P. penetrans* and Ptho*: P. thornei*.

**Figure 2 plants-10-00568-f002:**
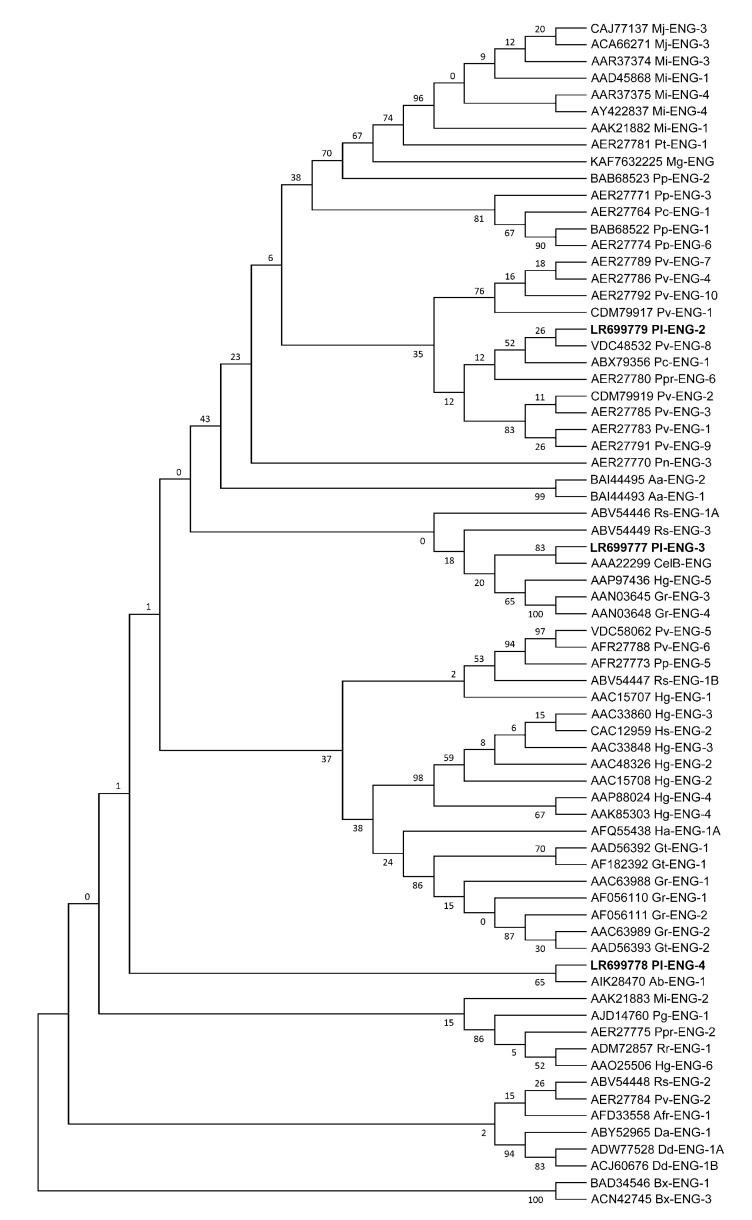
Phylogenetic relationships among the amino acid cellulase catalytic domains of plant parasitic nematodes. The Maximum Likelihood method was used to obtain a bootstrap consensus tree inferred from 1000 replicates.

**Figure 3 plants-10-00568-f003:**
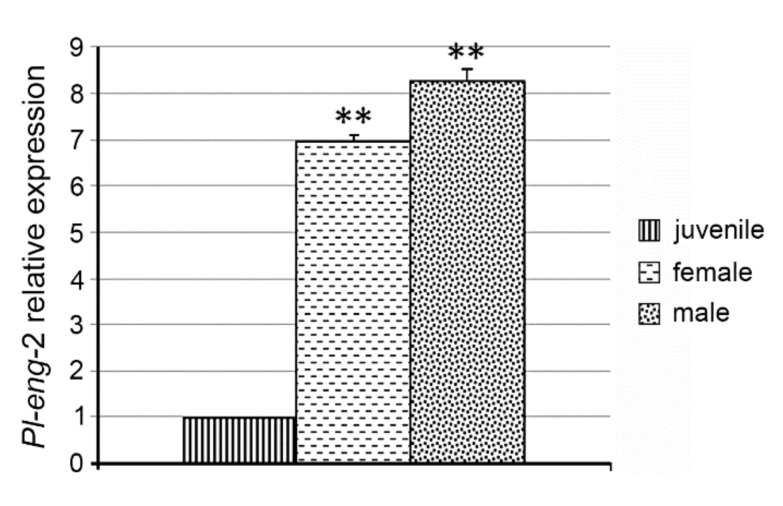
Expression of the *Pl-eng*-2 in juveniles (J2) and adult females and males. Bars indicate standard errors of the mean data (*n* = 3). Significant differences were found between J2 and the adult stages (** *p* < 0.01).

**Figure 4 plants-10-00568-f004:**
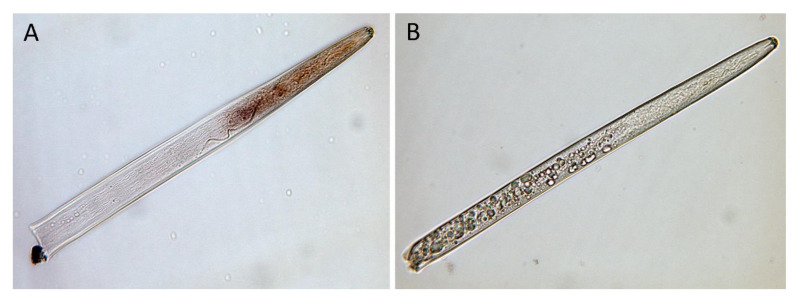
Localization of *Pl-eng*-2 in *P. loosi* by in-situ hybridization (**A**). No staining is observed with the sense probe (**B**).

**Figure 5 plants-10-00568-f005:**
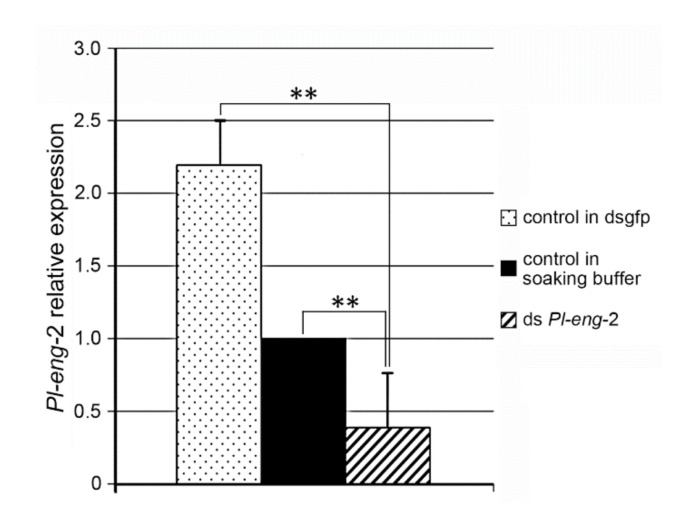
Relative expression level of *Pl-eng*-2 in nematodes after soaking in the green fluorescent protein gene dsRNA (*dsgfp*) control, in nematodes soaked in soaking buffer and in nematodes soaked in the target ds*Pl-eng*-2 by using real-time quantitative PCR. Bars indicate standard errors of the mean data (*n* = 3). Significant differences were found between the controls and treated nematodes (** *p* < 0.01).

**Figure 6 plants-10-00568-f006:**
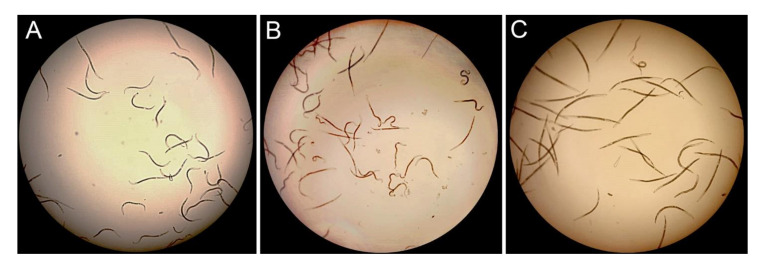
RNAi-mediated phenotypes of *P. loosi* nematodes treated with dsRNA *gfp* (**A**), untreated (**B**) and dsRNA *Pl-eng*-2 (**C**) after 24 h of soaking at 23 °C.

**Figure 7 plants-10-00568-f007:**
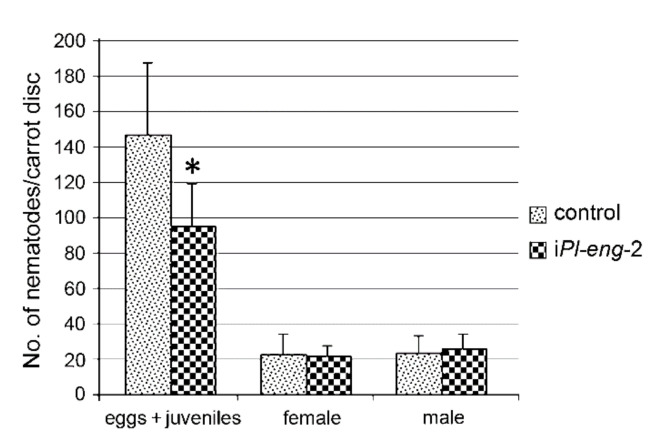
Number of eggs/juveniles and adult females and males recovered from carrot discs 45 days after inoculation with untreated and silenced nematodes. Bars indicate standard errors of the mean data (*n* = 3). Significant differences were found between the controls and treated nematodes (**p* < 0.05).

## Data Availability

The data presented in this study are openly available.

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
