# Peer review of "Molecular Characterization of Three B-1,4-Endoglucanase Genes in Pratylenchus loosi and Functional Analysis of Pl-eng-2 Gene"

_plants, 2021, doi:10.3390/plants10030568_

Round 1
Reviewer 1 Report
This is a well prepared manuscript presenting new information on "Molecular characterization of three B-1,4-endoglucanase genes in Pratylenchus loosi and functional analysis of Pl-eng-2 gene" worthy of publication. This study reports on the characterization and evolution of three ß-1,4-engoglucanase genes Pl-eng-2, Pl-eng-3 and Pl-eng-4. The gene structure of Pl-eng-2 was fully determined with the predicted signal peptide and devoid of the linker domain and carbohydrate-binding domain, while Pl-eng-3 and Pl-eng-4 were only partially sequenced. The transcription of Pl-eng-2 was localized in the secretory esophageal glands of all life stages, but it was upregulated in male and females stages. Exon/intron structures of Pl-eng-2, Pl-eng-3 and Pl-eng-4 confirmed that they resulted from gene duplication followed by sequence and gene structure diversification with loss of linker domain and carbohydrate-binding domain during evolution. Interesting to note was phylogenetic analysis further confirmed that nematode endoglucanases resulted from horizontal gene transfer of a bacterial gene as Pl-eng-3 showed sister relationships with CelB cellulase of Bacillus subtilis. In addition, silencing Pl-eng-2 by in vitro RNA interference, produced a 60% decrease of the transcript level. The reproductive ability of silenced P. loosi showed a 35% reduction of eggs and larval stages compared to untreated nematodes suggesting that this gene is involved in the early steps of invasion. The manuscript is recommended for publication in "Plants"
Author Response
We thank a lot referee 1 for the positive evaluation of our manuscript.
Reviewer 2 Report
The present study by Mirghasemi et al., reports on the charachterization and evolution of three β1-4-endoglucanase genes Pl-eng-2, Pl-eng-3 and Pl-eng-4 of the nematode Pratylenchus loosi infecting Tea plants in Iran. Through the study authors describe the gene structure of Pl-eng-2 which was fully determined and provide the partial sequences of additional two genes encoding for β1-4-endoglucanase, Pleng-3 and Pl-eng-4. Through their study the authors focus on expression analysis conducted by qRT-PCR and in situ hybridization only of Pl-eng-2 encoding gene. In silco analysis regards exon/intron position suggesting that these genes resulted from gene duplication followed by sequence and gene structure diversification with loss of linker domain and carbohydrate-binding domain during evolution. Function analysis of Pl-eng-2 using gene silencing methodology resulted in 60% reduction in Pl-eng-2 expression while reproduction ability of Pl-eng-2 silenced nematodes reduced by 35% compared with the control.
Authors provide a detailed function analysis of Pl-eng-2 of Pratylenchus loosi. However the results obtained following silencing are surprising in relation to the function of β1-4-endoglucanase in cell wall degradation. As motile inhibition and the phenotype observed in Fig. 6 doesn't correlate with expected function of β1-4-endoglucanase as a cell wall degradating enzyme. Moreover, Its phenotype following silencing, as observed by attenuation in viability, might lead to reduced amount of juveniles able to penetrate the carrot discs. This issue should be solved in order to correlate the reduced infection observed following Pl-eng-2 silencing. Author should add into the discussion more literature with information regards the phenotype that is observed following silencing of other cell wall degrading enzymes or ensure that all juveniles used for infection were viable similar to the control treatments.
Minor comments
*Abstract please correct endoglucanase
*This fragment was cloned
*Results section is broken A glycosyl….
In the Material and methods of the Expression analysis section, why expression analysis was not studied following exposure to plant (carrot discs) it might be that gene expression is regulated by plant signals
Through the Discussion it is written "The occurrence of several genes in Pratylenchus suggests that different eng genes are needed in Pratylenchus spp in order to contrast and to suppress the defense responses of the host plants during invasion and migration in the root…." While the function role of β1-4-endoglucanase enzymes in cell wall degradation and softening the root tissue its implication in suppression of plant defense is rather displaced.
Through the discussion "The conservation of intron position of Pl-eng-2 IN MOST ENGS of Pratylenchus suggests that this intron nay occur in the common ancestral gene "– this issue is repeated several time through the discussin and need to be mentioned once as it is redundant several times
What was found in P. vulnus, the expression in late stages ?
Expression pattern analysis please indicate that qRT-PCR analysis was conducted

Author Response
Replay to Referee 2
- We agree with referee 2 that Pl-eng-2 as β1-4-endoglucanase is a cell wall degradating enzyme. The phenotype “straight shape” observed in loosi after RNA silencing of Pl-eng-2 is due to the ingestion of dsRNA or entering the nematode body through other orifices during the incubation time. As reported by several authors, silenced genes can affect in different way the behavior and the phenotype of nematodes compared to nematode control (Fanelli et al., 2018; Iqbal et al., 2020; Fanelli et al. 2021). The straight shape of Pl-eng-2 dsRNA nematodes changed quickly after washing with water. The observation to the microscope revealed that all nematodes were viable and thus quickly used to incubate carrot discs.
2 Minor comments
*Abstract please correct endoglucanase
*This fragment was cloned
*Results section is broken A glycosyl…
We have directly included the corrections along the text.
3 The expression analyses were carried out on different life stages recovered from carrot discs and quickly stored at -80°C.
4 In literature are reported several effectors like endoglucanases that can interact with host proteins, in particular manipulating plant defense responses (Quentin et al., 2013; Vieira et al., 2018; Fanelli et al., 2019) In a previous paper on the roles of engs of P. vulnus, we described the endoglucanase Pv-eng-5 localized at intestine level demonstrated a direct relationship of the intestine with the environment and Pv-eng-5 may be involved in the plant defense evasion and basic defense against environmental toxins.The diversity of effectors highly abundant in the glands is likely to be related to the wide range of molecular functions that are required for penetration and invasion of host roots, detoxification, suppression of host defenses and many other unidentified functions (Rehman et al., 2016).
5 We have checked and deleted the redundant sentences in the discussion "The conservation of intron position of Pl-eng-2 that this intron nay occur in the common ancestral gene” as suggested by referee 2.
6 We have already found in P. vulnus that Pv-engs were expressed at high level in adult stages (females and males) as in P. loosi suggesting an active role of these stages in parasitism.
Reviewer 3 Report
1st question.
4.3. Genomic DNA amplification
Your statement is "Cycling conditions used were: an initial denaturation at 94°C for 2 min, followed by 45 cycles of denaturation..."
I think that 45 cycles is way too much in a PCR that you use PCR product for sequencing and continue with cloning procedure, considering the PCR mistakes
and artifacts. What is your thoughts? Are you using a special master mix (e.g. HIFI polymerase)?. Are there any SNIPs in you cloning sequencing results?
2nd question.
4.7. Expression pattern analysis
Your statement is "The relative expression among life stages was calculated by using the ΔΔCt method"
I do not see any indication of the number of individuals. I understand that you calibrate the difference between the stages with ΔΔCt method however even one individual probably could make the difference, am I right? How many individuals did you use for this test? collecting method?
3rd question.
2.6. Silencing of Pl-eng-2
Your statement is "No differences were observed in the number of males and females (Fig. 7)".
Why you have no significant differences in the adult numbers for this experiment. Were you expecting differences? If yes you should write soemthing on the discussion if not, why do you show this results?
Author Response
Reply to referee 3
1st question.
We agree with referee 3, we used 35 cycles , it was a mistyping (Fanelli et al.,2014. We sequenced different clones and we did not find sequence diversity among clones, otherwise we would have commented.
2nd question.
Transcript accumulation was calculated using the Comparative Ct method (ΔΔCt) with reference to expression of control.
We have added directly the number of specimens used to determine the expression pattern.
A different number of nematodes couldn’t make differences as we started from the same amount of male, female and J2 RNA, the same amount of cDNA and the amount of target is normalised to endogenous control 18S RNA and relative to the untreated sample (control) as calibrator.
3rd question.
First of all, we would like to stress that silencing by soaking is transient and also depends from the genes that you are going to silence. In our study, after 45 days (one life cycle for P. loosi) of incubation on carrot discs, we observed a decrease in reproduction (35%) of final population compared to the control population. The sum of the number of eggs, juveniles, females, and males was considered as the final nematode population. Then we demonstrated that this decrease is due to a reduced number of eggs and larval stages, while the ratio of females: males was comparable with that of the control population. Thus we have interpreted this result as the consequence of a reduced ability to feed and finally to reproduce.